# The Utility of Ultrasonography in the Diagnosis of Cervical Lymph Nodes after Chemoradiotherapy for Head and Neck Squamous Cell Carcinoma

**DOI:** 10.3390/medicina57050407

**Published:** 2021-04-23

**Authors:** Hoshino Terada, Yuzo Shimode, Madoka Furukawa, Yuichiro Sato, Nobuhiro Hanai

**Affiliations:** 1Department of Head and Neck Surgery, Aichi Cancer Center Hospital, 1-1 Kanokoden, Chikusa-ku, Nagoya, Aichi 464-868, Japan; hoshinoterada@aichi-cc.jp; 2Departments of Head and Neck Surgery, Kanazawa Medical University, 1-1 Daigaku, Uchinada cho, Kahoku gun, Ishikawa 920-0293, Japan; yuzo@kanazawa-med.ac.jp; 3Departments of Head and Neck Surgery, Kanagawa Cancer Center, 2-3-2 Nakao, Asahiku, Yokohama, Kanagawa 241-8515, Japan; madoka@yokohama.email.ne.jp; 4Departments of Head and Neck Surgery, Niigata Cancer Center Hospital, 2-15-3 Kawagishi cho, Chuo-ku, Niigata 951-8566, Japan; satox005@niigata-cc.jp

**Keywords:** ultrasonography, head and neck cancer, CRT, metastatic lymph node, PET-CT

## Abstract

*Background and Objectives*: There is evidence or consensus on the use of 18F-2-fluorodeoxyglucose-positron emission tomography with computed tomography (PET-CT) in evaluating the effects of treatment at 12 weeks after chemoradiotherapy for head and neck squamous cell carcinoma with cervical lymph node metastasis. However, the use of imaging to evaluate the effects of treatment within 12 weeks after chemoradiotherapy is controversial. The aim of this study was to evaluate the usefulness of ultrasonography in the diagnosis of lymph nodes metastasis after chemoradiotherapy according to the criteria of the “Evaluation of the effects of treatment on metastatic cervical lymph nodes using ultrasonography”, which evaluated lymph nodes metastasis based on size change and presence of degeneration. *Materials and methods*: This prospective study included 34 head and neck squamous cell carcinoma patients with cervical lymph nodes metastasis. Thirty-two patients who completed treatment were analyzed. Ultrasonography was performed at 4 and 8 weeks after chemoradiotherapy and we judged whether a favorable prognosis could be expected or whether additional treatments should be considered. Ultrasonography and PET-CT were performed at 12 weeks after chemoradiotherapy. Neck dissection was performed if residual disease was suspected based on the PET-CT findings. *Results*: The accuracy and negative predictive value of ultrasonography were 81.3% and 96.3%, respectively. According to the Ultrasonography findings, the size of lymph nodes metastasis after chemoradiotherapy was significantly smaller than those before chemoradiotherapy (*p* < 0.05). The fluid and blood flow of lymph nodes metastasis showed a significantly reduced at 12 weeks after chemoradiotherapy (*p* < 0.05, *p* < 0.05, respectively). The echo density significantly changed from low to high echoic density after chemoradiotherapy (*p* < 0.05). *Conclusions*: Ultrasonography was useful for evaluating cervical lymph nodes metastasis after chemoradiotherapy for head and neck squamous cell carcinoma.

## 1. Introduction

Recently, chemoradiotherapy (CRT), which is performed for the purpose of organ preservation, has become standard of care for locally advanced head and neck squamous cell carcinoma [1,2]. The presence of metastatic neck nodes is associated with a worse prognosis [3]. If it is suggested the presence of residual lymph node metastasis after CRT, neck dissection is performed. However, we sometimes experience cases in which there is no pathological remnant after neck dissection. Mehhanne et al. reported that the rates of all complications and severe complications in a patients undergoing planned neck dissection were 38% and 26%, respectively [4]. Unnecessary or futile treatment should always be avoided.

The negative predictive value of 18F-2-fluorodeoxyglucose-positron emission to-mography with computed tomography (PET-CT) at least 12 weeks after CRT is reportedto be high. Nelissen et al. reported that negative predictive value, sensitivity and specificity of PET-CT were 92%, 73% and 83%, respectively [5]. PET-CT performed before 3 months after CRT is also reported to have higher false-negative rates and higher false-positive rates [5,6]. For the evaluation of treatment effect, the National Comprehensive Cancer Network (NCCN) guidelines recommend that PET-CT be performed a minimum of 12 weeks after CRT [4,7]. However, PET-CT is considered to be associated with higher cost in comparison to enhanced computed tomography (CT) or magnetic resonance imaging (MRI), and these technologies are not evenly distributed at all facilities. Further, there are patients who require treatment before 12 weeks after CRT because of tumor growth. The NCCN guidelines recommend that CT or MRI be performed at 4–8 weeks after CRT in cases in which a clinical assessment raises concerns regarding the possibility of a residual tumor [7]. However, the sensitivity, specificity, and accuracy of CT and/or MRI at 4–8 weeks after CRT were reported to be insufficient (60–66.7%, 67–73.8% and 60–72.8% [8,9]). The modality that is most appropriate for the assessment of lymph nodes within 12 weeks after CRT remains controversial.

Ultrasonography (US) is useful because it is a simple, non-invasive imaging tech-nique. Grayscale ultrasonography can be used to evaluate the distribution and morphology of the lymph nodes, whereas Doppler ultrasonography can be used to evaluate the distribution of intranodal vascularity [10,11,12]. It is particularly important to evaluate whether metastatic lymph nodes will degenerate before 12 weeks after CRT because the performance of neck dissection should be considered if lymph nodes are enlarged. The diagnosis of residual lymph nodes metastasis by CT has been considered to show poor specificity [8], because the diagnosis is mainly determined based on size, rather than qualitative information. However, progress in imaging technology has allowed the US to be applied in the qualitative evaluation of lymph nodes with the use of Doppler blood flow and tissue elasticity. US is also advantageous in that it can be performed repeatedly, even monthly, to evaluate the degeneration of metastatic lymph nodes. To the best of our knowledge, no reports have evaluated the degeneration of metastatic lymph nodes after CRT using ultrasonography based on the established diagnostic criteria for head and neck squamous cell carcinoma.

In the present study, we aimed to evaluate the usefulness of US in the diagnosis of cervical lymph nodes after CRT for head and neck squamous cell carcinoma. The evaluation was performed according to the criteria of the “Evaluation of the effects of treatment on metastatic cervical lymph nodes using ultrasonography”, which were proposed by Furukawa MK, who belongs to our Head and Neck Ultrasound Study Group [13].

## 2. Materials and Methods

### 2.1. Patients

Between 2014 and 2017, 34 head and neck squamous cell carcinoma patients with cervical lymph node metastasis were enrolled in this prospective study. Two patients did not finish the protocol treatment because of adverse effects. Thus, a total of 32 patients were evaluable. The patients were enrolled across 4 hospitals belonging to the Head and Neck Ultrasound Study Group in Japan: Aichi Cancer Center Hospital, Kanazawa Medical University, Kanagawa Cancer Center, and Niigata Cancer Center Hospital. We included patients who received CRT. Patients who received induction chemotherapy followed by radiotherapy or bioradiotherapy (BRT) with cetuximab were also included. The dose of radiation to the primary tumor and metastatic lymph nodes was at least 60 Gy. The dose of radiation to the neck lesion, as a prophylactic dose, was at least 40 Gy. Clinical lymph node metastasis was diagnosed according to preoperative imaging, including routine CT and/or MRI. If possible, PET-CT was performed with the intention of diagnosing the neck status. The tumors were staged according to the seventh edition of TNM Classification of Malignant Tumors, published in affiliation with International Union Against Cancer (UICC). We excluded patients with distant metastasis. We also excluded patients with N3 (Metastasis in a lymph node more than 6 cm in greatest dimension) because cervical lymph node metastasis was too bulky to visualize and evaluate by ultrasound. We also excluded patients who were prescribed adjuvant chemotherapy after definitive CRT. All subjects gave their informed consent for inclusion before participation in the present study. The study was conducted in accordance with the Declaration of Helsinki, and the protocol was approved by the Institutional Review Boards (Project identification code; 2014-1-035, dates of approval; 21 August 2014).

### 2.2. Follow-Up

Figure 1 shows a schematic diagram of the protocol of this study. We performed US at 4 and 8 weeks after CRT. In cases in which we judged that additional treatment should be considered, neck dissection was performed. At 12 weeks after CRT, US and PET-CT was performed. We performed neck dissection and primary dissection as necessary it was judged, based on the PET-CT results, that additional treatment should be considered. In operative cases, we evaluated the lymph nodes that were subjected to a pathological examination. In observation cases, patients were observed for at least six months to detect recurrence at the lymph nodes. Patients were followed for 2 years after CRT.

### 2.3. Ultrasound Evaluation Methods

The size of the lymph nodes was measured using three-dimensional ultrasonography. One lymph node was investigated per patient. In cases involving multiple lymph node metastases, the lymph node with the largest maximum diameter was targeted. The change in size in a bi-dimensional measurement (thickness × minor axis) was used to evaluate the treatment effects (Figure 2).

Grade 1: Lymph node metastasis disappeared.Grade 2: Metastatic lymph nodes returned to a normal structure.Grade 3: Metastatic lymph nodes regressed ≥50%.Grade 4: Metastatic lymph nodes regressed <50%.Grade 5: Metastatic lymph node was enlarged.

Figure 3 shows the criteria for the “evaluation of treatment effects on cervical metastatic lymph nodes using ultrasonography”.

A: Disappeared or scarred, indicating the effacement of the metastatic lymph nodes or that only scars remained.B: Regression, indicating that the metastatic lymph nodes had regressed and remained (bidirectional regression rate: ≥50%).C: Unchanged, indicating that the metastatic lymph nodes had not regressed (bidirectional regression rate: <50%).D: Worsening, indicating that the metastatic lymph nodes were enlarged

Metastatic lymph nodes classified as B or C were further classified according to the presence or absence of degeneration. We decided that metastatic lymph nodes that had lost blood flow were degenerated. We also evaluated echo density (low-iso-high), homogeneity (homogeneity or inhomogeneity) and whether metastatic lymph nodes showed fluid or blood flow.

### 2.4. Statistical Analyses

The accuracy of US was calculated by the following formula: ((Number of cases in which the US assessment and pathological diagnosis match) + (Number of cases in which US assessment and the presence or absence of neck failure match)) divided by (Number of cases evaluated by US). At 12 weeks after CRT, a diagnosis was judged as “correct” when the US assessment and the presence or absence of pathologically diagnosed residual lymph node metastasis matched in patients undergoing neck dissection. In patients undergoing observation, neck failure was defined as lymph node enlargement within 6 months after CRT.

The accuracy of PET-CT was calculated as follows: ((Number of cases in which the PET-CT assessment and pathological diagnosis match) + (Number of cases in which PET-CT assessment and the presence or absence of neck failure match)) divided by (Number of cases evaluated by PET-CT). At 12 weeks after CRT, a diagnosis was judged as correct when the assessment of PET-CT and the presence or absence of pathologically diagnosed residual lymph node metastasis matched in patients undergoing neck dissection.

In patients undergoing observation, neck failure was defined as lymph node enlargement during 6 months after CRT. The sensitivity, specificity, positive predictive value and negative predictive value were calculated using two by two table.

The US findings included fluid, blood flow, echo density and homogeneity. The correlation of these US findings before CRT with those at 12 weeks after CRT were analyzed using Wilcoxon’s signed-rank test. The correlation of the size at 4 weeks after CRT with that at 12 weeks after CRT was analyzed using the Wilcoxon’s signed-rank test.

Statistical analyses were performed using the EZR software program (Saitama Medical Center, Jichii Medical University, Saitama, Japan) [14], which is a graphical user interface for R (The R Foundation for Statistical Computing, Vienna, Austria).

## 3. Results

The patient characteristics are shown in Table 1. The median age was 64.5 years (range, 46–77). The median radiation dose was 70 Gy (range, 60–70) for the primary tumor and metastatic lymph nodes. In total, 23 patients had received concurrent CRT with intravenous cisplatin every 3 weeks (80 or 100 mg/m^2^), cisplatin every week (30 or 40 mg/m^2^) or cisplatin (70 mg/m^2^) and 5-fluorouracil (700 mg/m^2^). Among 23 patients, 10 patients had undergone induction chemotherapy with TPF; docetaxel (75 mg/m^2^, day 1), cisplatin (75 mg/m^2^, day 1) and 5-fluorouracil (750 mg/ m^2^, day 1–5) or PF; cisplatin (80 mg/m^2^, day 1) and 5-fluorouracil (800 mg/m^2^, day 1–4). Three patients underwent induction chemotherapy with PF followed by radiotherapy. Six patients had undergone concurrent bioradiotherapy with cetuximab (400–250 mg/m^2^). Among these 6 patients, 3 underwent induction chemotherapy with TPF or PF.

A flowchart of the clinical course of the32 examined patients is shown in Figure 4. At 4 or 8 weeks after CRT, a favorable prognosis was predicted based on US findings in 27 patients. Among 27 patients, 3 patients were diagnosed with residual lymph nodes metastasis and 1 patient was diagnosed with a residual primary tumor based on PET-CT at 12 weeks after CRT. Two patients underwent neck dissection, only 1 patient had a pathologically positive lymph node. One patient, who was diagnosed with a residual primary tumor based on PET-CT at 12 weeks after CRT, underwent primary resection and neck dissection dissection. A pathological examination revealed no residual lesion. One patient was observed. The patient developed recurrent disease 7 months later and died. In five patients, it was judged that additional treatment should be considered based on the US findings at 4 or 8 weeks after CRT. Among these 5 patients, 2 were diagnosed with residual lymph nodes metastasis based on PET-CT at 12 weeks after CRT.

One patient was treated with neck dissection, and another was treated with fine-needle aspiration cytology. In both cases, the lymph nodes were pathologically negative. Although three patients had no residual lesion on PET-CT at 12 weeks, one patient under-went neck dissection based on an US assessment. However, the pathological examination revealed no residual lymph nodes metastasis.

The accuracy, specificity and negative predictive value of US were 81.3%, 83.9% and 96.3%, respectively. The accuracy, specificity and negative predictive value of PET-CT were 87.5%, 87.1% and 100%, respectively (Table 2).

According to the US findings, the size of metastatic lymph nodes, fluid, echo density and blood flow showed significant differences between before CRT and 12 weeks after CRT. At 12 weeks after CRT, the classification of size were as follows: grade 1 or grade 2 (*n* = 12), grade 3 (*n* = 18), and grade 4 (*n* = 1). One patient was not evaluated because we performed surgery without waiting for 12 weeks after CRT. Although the difference between the size of lymph nodes metastasis at 4 weeks and those at 8 weeks after CRT did not reach statistical significant (*p* = 1), there was a significant difference between the size of lymph nodes metastasis at 8 weeks after CRT and those at 12 weeks after CRT (*p* = 0.01). In addition, there was a significant difference between the size of lymph nodes metastasis at 4 weeks after CRT and those at 12 weeks after CRT (*p* = 0.006).

Although the lymph nodes metastasis was gradually decreased during and after treatment, the changes were particularly significant between at 8 weeks and at 12 weeks. The size change of lymph node metastasis mainly showed the following two patterns: Grade 4 at 4 weeks after CRT, then reduction to grade 3 at 12 weeks after CRT. And Grade 3 at 4 weeks after CRT, then reduction to grade 2 at 12 weeks after CRT.

The fluid level of metastatic lymph nodes was significantly reduced at 12 weeks after CRT (*p* < 0.05). The echo density significantly changed from low to high echoic density (*p* < 0.05). The blood flow of metastatic lymph nodes also showed a significant decrease (*p* < 0.05) (Figure 5).

The homogeneity of metastatic lymph nodes before and after CRT did not differ to a statistically significant extent. Ultrasonographic appearance of metastatic lymph nodes at 4, 8, 12 weeks were summarized in Figure 6.

## 4. Discussion

In this study, the accuracy and negative predictive value of US were 81.3% and 96.3%, respectively and the accuracy and negative predictive value of PET-CT were 87.5% and 100%, respectively. The sensitivity and positive predictive value of US were 0%. This was because the true positive and false negative were zero and one, respectively in two by two table. The reason was that there was only one patient with residual tumor at 6 months after CRT, who was evaluated by US as a favorable prognosis can be expected.

Our previous study revealed that the accuracy of CT or MRI in the diagnosis of residual lymph node metastasis at 6 weeks after the completion of CRT was 60% (specificity, 67%; negative predictive value, 77% [9]. US was less accurate than PET-CT but more accurate than CT.

Nishimura et al. reported that the sensitivity, specificity and accuracy of CT and/or MRI at 4–8 weeks after CRT were 66.7%, 73.8% and 72.8%, respectively. They concluded that CT and/or MRI were inadequate for evaluating lymph node metastasis after CRT. Further, they reported that the accuracy, specificity and negative predictive value of US at 8 weeks after CRT were 82.4%, 82.8% and 96.0%, respectively [8]. Some studies reported that the PET-CT findings at 12 weeks after CRT had high negative predictive value. However, PET-CT performed within 3 months had higher-false negative and false-positive rates [5,6]. There is no consensus on how to evaluate lymph nodes metastasis within 12 weeks after CRT.

US is less invasive and cheaper to perform in comparison to CT and can be performed repeatedly. In addition, we can measure the size of lymph nodes, which is an important aspect in the assessment of residual lymph nodes metastasis by CT. Thus, based on the merits of US, we are of the opinion that it is superior to CT for this task. We can observe metastatic lymph nodes using US during treatment. Using this approach, when a patient is diagnosed with grade 1 or 2 metastatic lymph nodes based on US, treatment may be initiated with the expectation of a favorable prognosis without waiting for the PET-CT results.

To improve the accuracy of US, it is necessary to clarify the characteristics of other US findings. In this study, fluid, echo density and blood flow showed significant differences before and after CRT. When we evaluate the presence or absence of metastatic lymph node degeneration, we should observe not only the reduction of flood flow but also the reduction of fluid or the change from low to high echoic density. Moreover, we should consider adding the absolute value of the lymph node size to our flow chart as one of the diagnostic criteria. We devised new diagnostic criteria for “Evaluation of the effects of treatment on metastatic cervical lymph nodes using ultrasonography” (Figure 7). The findings of metastatic lymph nodes using US include hypoechoic density, fluid due to intranodal necrosis and blood flow from areas other than the hilum [12,13]. Findings of degeneration include a change from low to high echoic density and the disappearance of fluid and blood flow. In the future, it will be necessary to carry out further studies using new criteria.

The present study was associated with several limitations. First, the number of patients was relatively small. Second, the quality of US was dependent on the sonographer’s skill, and not all sonographers are familiar with the evaluation of metastatic lymph nodes.

Third, the absence of US at the begin and/or end of chemoradiotherapy reduce ability of the study achieve conclusions. Fourth, limiting the evaluation to only one lymph node per each patient is an important limitation. When evaluating metastatic lymph nodes if there is effect of treatment, it is often done for the largest lymph node. That is why this study limited it to only one lymph node per each patient. However, in patients with multiple metastatic nodes, some of them may regress, while others may enlarge. Fifth, the problem with this technology is that is does not allow evaluation of the response of the primary tumor, so it should be complemented with one of the other imaging techniques, and its usefulness is therefore limited. Sixth, the tumors were staged according to the seventh edition of TNM Classification of Malignant Tumors, published in affiliation with UICC.In the future, it is imperative to conduct a trial using the eighth edition of TNM Classification of Malignant Tumors.

## 5. Conclusions

This is the first study to evaluate the effects of treatment after CRT using US criteria. This study revealed changes in metastatic lymph nodes before and after CRT using US. In the present study, US was less accurate than PET-CT, however, it is possible that the accuracy will improve with the newly proposed diagnostic criteria.

## Figures and Tables

**Figure 1 medicina-57-00407-f001:**
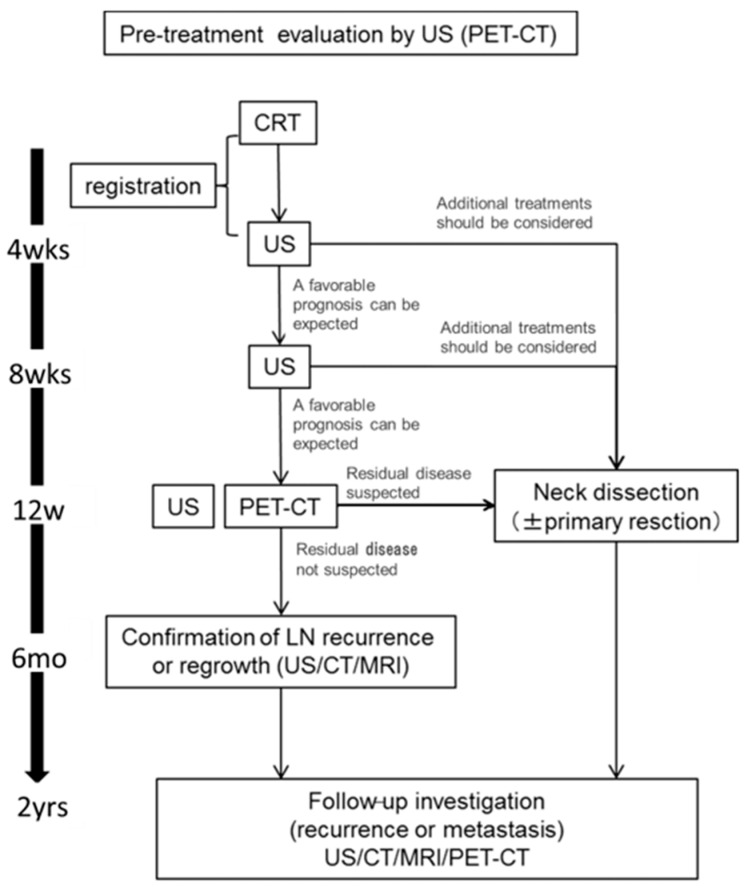
A flowchart of the study design. Patients underwent US examinations at 4, 8, 12 weeks after CRT and PET-CT at 12 weeks after CRT. If it was deemed that additional treatment should be considered based on the suspicion of residual disease, patients underwent neck dissection. At 6 months after CRT, the recurrence of lymph nodes was confirmed by imaging. The duration of follow-up was 2 years after CRT. CRT, chemoradiotherapy; US, Ultrasonography; CT, enhanced computed tomography; MRI, magnetic resonance imaging; PET-CT, 18F-2-fluorodeoxyglucose-positron emission tomography with computed tomography.

**Figure 2 medicina-57-00407-f002:**
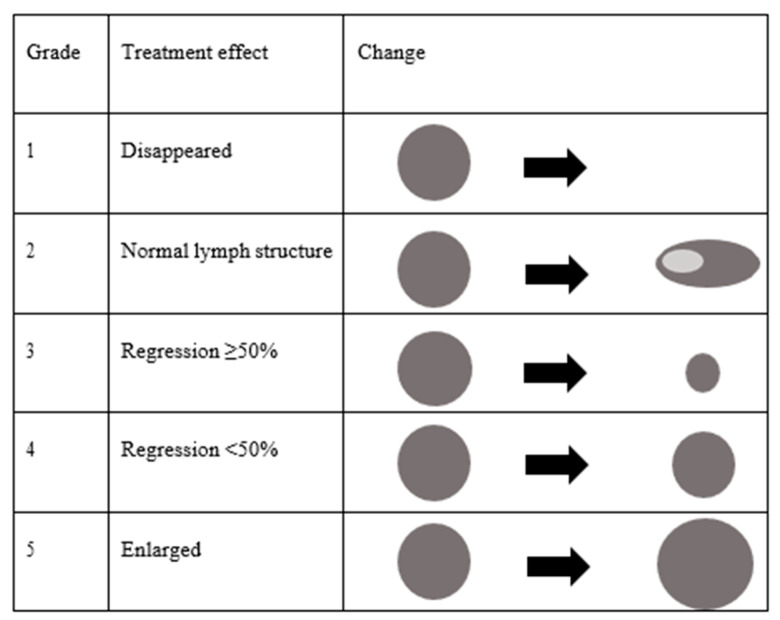
Classification of size change on ultrasonography after chemoradiotherapy.

**Figure 3 medicina-57-00407-f003:**
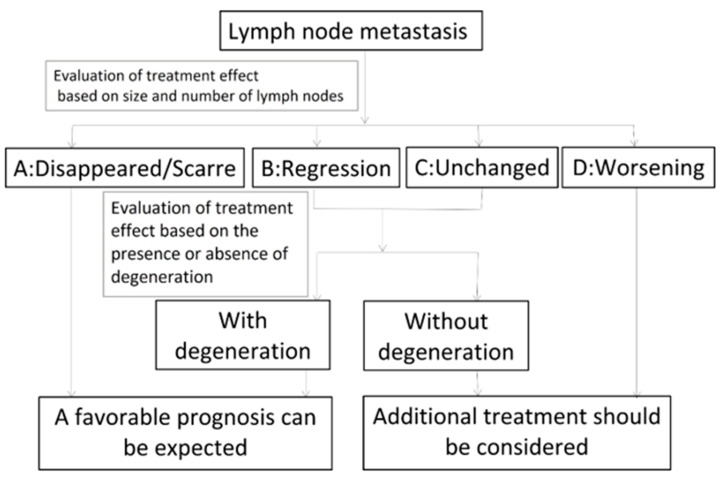
Evaluation of the effects of treatment on metastatic cervical lymph nodes using ultrasonography.

**Figure 4 medicina-57-00407-f004:**
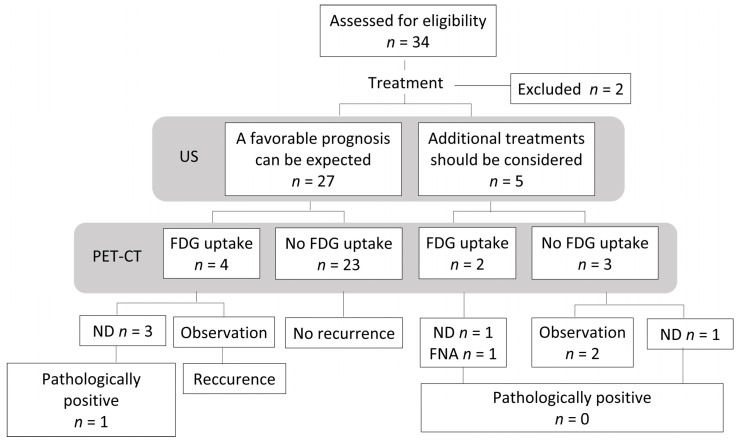
Flowchart of the study population. US, Ultrasonography; PET-CT, 18F-2-fluorodeoxyglucose-positron emission tomography with computed tomography; FDG, fluorodeoxyglucose; ND, neck dissection; FNA, fine needle aspiration cytology.

**Figure 5 medicina-57-00407-f005:**
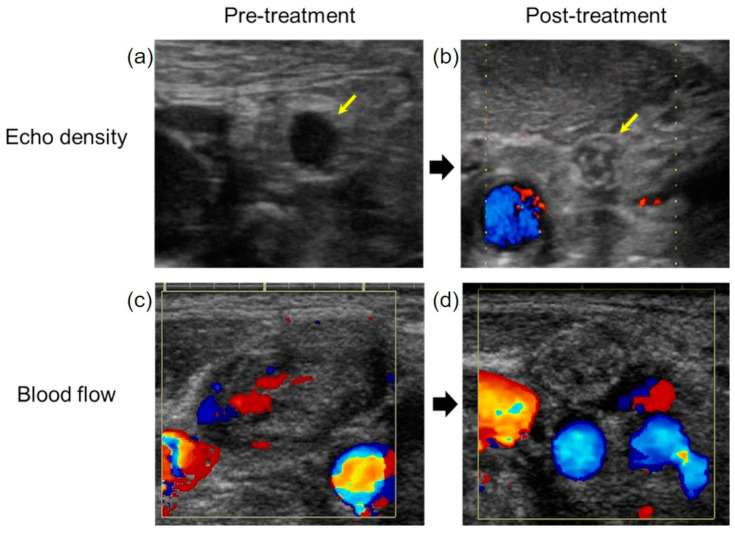
The evaluation of the treatment effect on metastatic lymph nodes. (**a**) The echo density of the metastatic lymph node was low before treatment. (**b**) The echo density changed to high after treatment (yellow arrow). (**c**) The blood flow of metastatic lymph node observed before treatment. (**d**) The blood flow to the metastatic lymph nodes disappeared after CRT.

**Figure 6 medicina-57-00407-f006:**
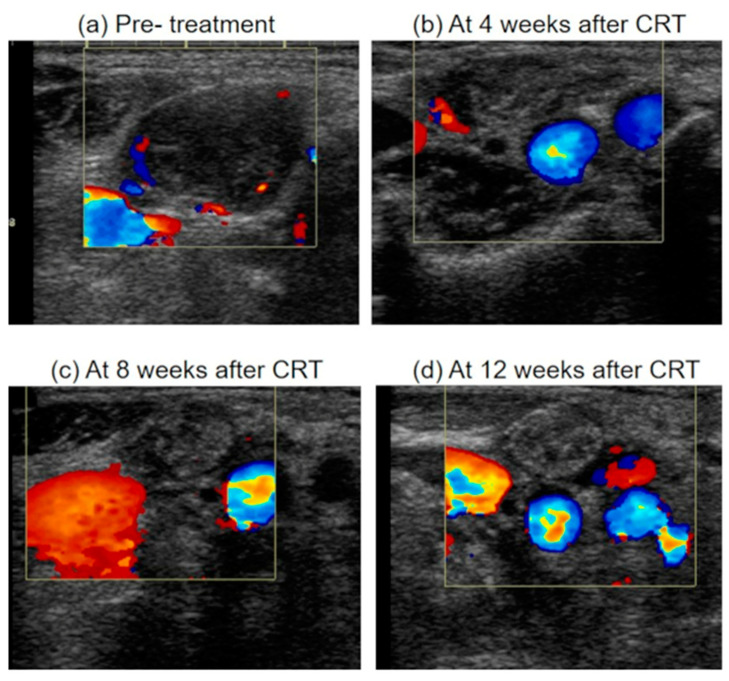
Ultrasonographic appearance of metastatic lymph node after chemoradiotherapy. (**a**) A metastatic lymph node before treatment, which was enlarged, inhomogeneity, hypoechoic and with blood flow from areas other than the hilum. (**b**) The metastatic lymph node regressed <50% (Grade 4) at 4 weeks after CRT. The echo density of lymph node was isoechoic. The blood flow from areas other than the hilum was decreased (**c**) The metastatic lymph node regressed ≥50% (Grade 3) at 8 weeks after CRT. The blood flow from areas other than the hilum was further decreased. (**d**) The metastatic lymph node regressed ≥50% (Grade 3) at 12 weeks after CRT. The blood flow from areas other than the hilum was disappeared. And the echo density of lymph node was high. CRT, chemoradiotherapy.

**Figure 7 medicina-57-00407-f007:**
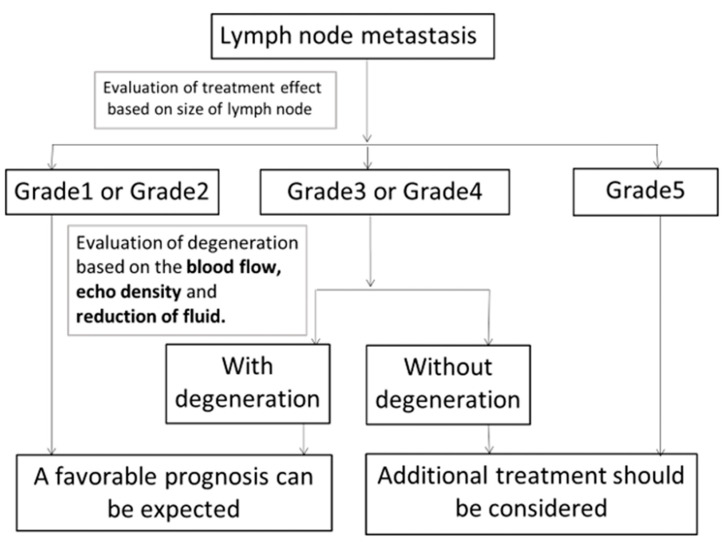
New diagnostic criteria for “Evaluation of the effects of treatment on metastatic cervical lymph nodes using ultrasonography”. After CRT, the effects of treatment on metastatic lymph nodes were evaluated based on size change. Patients with grade 1 or 2 lymph nodes were expected to have a favorable prognosis. Additional treatment should be considered for patients with grade 5 lymph nodes. Grade 3 or 4 lymph nodes were evaluated further to determine whether they showed degeneration based on the disappearance of blood flow, change of echo density to high and the disappearance of fluid. Patients with lymph node degeneration were expected to show a favorable prognosis. Additional treatment should be considered for patients with lymph nodes without degeneration.

**Table 1 medicina-57-00407-t001:** Patient characteristics (*n* = 32).

Characteristics		Number
Age	Median (range)	64.5 (46–77)
Sex	Male	29
	Female	3
Primary site	Nasopharynx	2
	Oropharynx	18
	Hypopharynx	10
	Larynx	2
Clinical T classification	T1	2
	T2	14
	T3	5
	T4	11
Clinical N classification	N1	4
	N2a	2
	N2b	15
	N2c	11
Induction chemotherapy	Presence	16
	Absence	16
CRT/RT		29/3
Chemotherapy	triweekly CDDP	11
	weekly CDDP	11
	CDDP + 5-FU	1
Biotherapy	Cetuximab	6

T, tumor; N, node; CRT, chemoradiotherapy; RT, radiotherapy; CDDP, cisplatin; 5-FU, 5-fluorouracil. N2a, Metastasis in a single ipsilateral lymph node, more than 3 cm but not more than 6 cm in greatest dimension; N2b; Metastasis in multiple ipsilateral lymph nodes, none more than 6 cm in greatest dimension; N2c; Metastasis in bilateral or contralateral lymph nodes, none more than 6 cm in greatest dimension.

**Table 2 medicina-57-00407-t002:** Sensitivity, specificity, positive predictive value, negative predictive value and accuracy of US and PET-CT.

	Sensitivity (%)	Specificity (%)	PPV (%)	NPV (%)	Accuracy (%)
US	0	83.9	0	96.3	81.3
PET-CT	100	87.1	20	100	87.5

PPV, positive predictive value; NPV, negative predictive value; US, ultrasonography; PET-CT, 18F-2-fluorodeoxyglucose-positron emission tomography with computed tomography.

## Data Availability

The data presented in this study are available on request from the corresponding author. The data are not publicly available due to privacy.

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
