# Peer review of "The Utility of Ultrasonography in the Diagnosis of Cervical Lymph Nodes after Chemoradiotherapy for Head and Neck Squamous Cell Carcinoma"

_medicina, 2021, doi:10.3390/medicina57050407_

Round 1

Reviewer 1 Report

This is an interesting study about ultrasonography in the diagnosis of cervical lymph nodes after chemoradiotherapy for head and neck squamous cell carcinoma. The authors analyzed the accuracy of ultrasonography in detecting residual disease after treatment.

All the acronyms should be explained at their first appearance, also in the abstract.

In the abstract, please rewrite the sentence about aims of the study without referring to whom proposed the study. Moreover, avoid the formula for accuracy in the abstract. Results must be implemented in the abstract.

In the Introduction section, the authors should report negative predictive value, sensibility and specificity of PET-CT.

The authors excluded patients with distant metastasis or with cervical lymph node metastasis that was too bulky to visualize and evaluate with ultrasound. Please better specify. Were N3 patients excluded?

The absence of ultrasonography at the begin and/or end of chemoradiotherapy reduce the ability of the study to achieve valid conclusions.

Furthermore, limiting the evaluation to only one lymph node per each patient is an important limitation of the study. Indeed, in patients with multiple metastatic nodes, some of them may regress, while others may enlarge.

The authors described two methods for evaluation, a grading system from 1 to 5, and criteria from 1 to 4, that are slight different. I think that using a unique system of classification may be more easy to understand.

How was it possible that sensitivity and positive predictive value were 0% for ultrasonography (table 2)?

Ultrasonographic appearance of nodes at 4, 8 and 12 weeks should be better summarized in the Results section. A table or graphic may help the reader.

There are several grammatical errors. The authors should submit the text to a native English speaker to improve readability.

Reviewer 2 Report

This paper analyzes the usefulness of ultrasonography to evaluate the response of lymph node metastases in patients with head and neck squamous cell carcinoma treated with chemoradiotherapy. Although the number of patients included is small, the results are interesting and suggest that ultrasonography can be used as an alternative to other imaging techniques to evaluate this response, with a higher accuracy than CT and MRI, but lower than the gold standard PET-CT.  The problem with this technology is that it does not allow evaluation of the response of the primary tumor, so it should be complemented with one of the other imaging techniques, and its usefulness is therefore limited. This should be explicitly mentioned in the manuscript.
Minor comments:
- Why are cases with bulky metastases excluded and what is meant by bulky metastases, N3a?
- In the description of how PET-CT accuracy is evaluated (page 6) there is an error in the denominator: where it says US, it should say PET-CT. 

Round 2

Reviewer 1 Report

Thank you for imrpoving the paper.

Author Response

Thank you very much for your insightful comments.